# Extent and nature of duplication in PROSPERO using COVID-19-related registrations: a retrospective investigation and survey

Lucy Beresford  , Ruth Walker, Lesley Stewart

Centre for Review and Dissemination, University of York, York, UK

**Correspondence to**
Ms Ruth Walker;
ruth.walker@york.ac.uk

## ABSTRACT

**Objectives** During COVID-19, the International Prospective Register of Systematic Reviews (PROSPERO) experienced a surge in registrations for COVID-19-related systematic reviews, and duplication of research questions became apparent. Duplication can waste funding, time and research effort and make policy making more difficult.

This project explored the extent of and reasons for duplication of COVID-19-related systematic review registrations in PROSPERO during the pandemic.

**Design** Retrospective analysis of COVID-19-related registrations in PROSPERO, and a qualitative survey.

**Setting** PROSPERO was searched for registrations related to four COVID-19 research areas: epidemiology, rehabilitation, transmission and treatments.

**Methods** Records identified were compared using Population, Intervention/Exposure, Comparator, Outcome, Study Design (PICOS) elements of PROSPERO registration forms. Registrations with similar or identical PICOS were evaluated further as 'duplicates'.

Authors of 'duplicate' registrations were invited to complete a survey asking whether they searched PROSPERO prior to registration, identified similar reviews and, if so, why they continued with their review.

**Results** 1054 COVID-19 reviews were registered between March 2020 and January 2021, of which 138 were submitted when at least one similar protocol was already registered in PROSPERO. Duplication was greatest in reviews of COVID-19 treatments; for example, there were 14 similar reviews evaluating the efficacy of hydroxychloroquine.

From 138 authors invited to take part in the survey, we received 41 responses. Most respondents said that they identified similar reviews when they searched PROSPERO prior to registration. Main reasons given for 'duplication' were differences in PICOS or planned analyses (n=13), poor quality of previous registrations (n=2) and the need to update evidence (n=3).

**Conclusions** This research highlights that registration of similar and duplicate systematic reviews related to COVID-19 in PROSPERO occurred frequently. Awareness of research waste is required, and initial checking for similar reviews should be embedded within good review practice.

## STRENGTHS AND LIMITATIONS OF THIS STUDY

⇒ As this research project was conducted by members of the International Prospective Register of Systematic Reviews (PROSPERO) team, we were able to access data that are not publicly available—specifically whether authors identified other similar reviews prior to registration. Therefore, our work provides a unique insight into the nature of duplication in PROSPERO.

⇒ The inclusion of a survey provided further insight into the nature of duplication in PROSPERO, which has allowed us to offer specific improvements for protocol registration in PROSPERO, specifically at a time where cooperation and collaboration between researchers were pertinent given the public health crisis.

⇒ We received a relatively low response to our survey (30%). Therefore, while it provides useful insights into the reasons for duplication, it may not be representative of all those who registered 'duplicate' reviews.

⇒ Our study was limited by available resource and was only able to explore the extent and nature of duplication from March 2020 until January 2021. Therefore, key milestones in the pandemic (such as the global roll-out of vaccinations, or the availability of antivirals) that could have impacted the extent of duplication were not explored.

⇒ For the same reason we were only able to evaluate the extent and nature of duplication in four of the 17 COVID-19-related topic areas. Although we attempted to choose a broad range of topics, the extent of duplication could differ in other COVID-19 research areas

## INTRODUCTION

During the COVID-19 pandemic, a significant volume of medical research has been conducted to better understand the virus, its biology and associated disease.[1] Large numbers of systematic reviews have synthesised the emerging evidence.[2] It is best practice to register systematic reviews at the protocol stage to help avoid unplanned duplication

and to enable comparison between methods reported in the final publication and that which was planned in the protocol.[3] This can be done in registries such as the International Prospective Register of Systematic Reviews (PROSPERO) protocols, funded by the National Institute for Health and Care Research in the UK. PROSPERO was launched in 2011 with the intention of increasing transparency, limiting unintended duplication and reducing the risk of bias in systematic reviews.[4–6]

In the 10 years since its launch in 2011, prospective registration of systematic reviews has become routine and the importance of this acknowledged widely. At launch, PROSPERO anticipated registration of around 2000 systematic reviews per year. In 2020, a total of 40 639 reviews were registered, and PROSPERO now includes over 125 000 registration records. PROSPERO has global reach, with registrations originating from 194 countries and territories in 2020. In March 2020, PROSPERO prioritised registration of review protocols related to COVID-19. Our intention was to support the global pandemic research endeavour by making information about planned and ongoing reviews public as quickly as possible and providing researchers and policy makers with an overview of the evidence synthesis pipeline. This would also help reduce unintended duplication by enabling authors to check whether similar reviews already existed before embarking on a new one, based on up-to-date information (COVID-19 registrations are published within 24 hours of submission from Monday to Friday). Submissions are registered in PROSPERO through the process described in box 1.

---

### Box 1    The registration process in PROSPERO

⇒ Registration in the International Prospective Register of Systematic Reviews (PROSPERO) involves the submission and publication of key information about the design and conduct of a systematic review.

⇒ Authors should register their systematic review once they have designed their review protocol but before they begin data extraction.

⇒ Prior to submission, authors answer a series of 'triage questions' to check whether their review is eligible for inclusion.

⇒ Previously, applications were checked by the PROSPERO team to ensure that they were in scope and that the required data had been provided. Since March 2020, to enable rapid processing of COVID-19 registrations, some applications have been published automatically following simple automated checks. No quality assessment or peer review is involved. Records are published on an open access electronic database.

⇒ Registration information can be amended by authors should plans change. These changes are published, and a date-stamped audit trail of previous versions made available in the public record.

⇒ Registration records are permanent, and authors are asked to provide links to subsequent reports and publications.

⇒ PROSPERO assigns each registered review a unique registration number. This number should be cited in publications.

⇒ Peer reviewers and readers can compare what was reported for the review with what was intended in the review protocol.

---

To aid identification of reviews on topics of key relevance to pandemic research, a single click search by broad COVID-19 research categories was implemented on the PROSPERO home page. The research categories have been regularly updated as new topics emerge. PROSPERO encourages authors not to duplicate existing systematic reviews but recognises that duplication may be intended with good reason (eg, the existing review is out of date or used suboptimal methods). As such, when registering, users are asked to indicate whether their review is (1) not similar to an existing review, (2) sufficiently different or (3) similar to an existing review, but repetition is needed, as part of a series of eligibility questions before completing the registration form.

Despite encouragement to check PROSPERO for similar systematic reviews and to not duplicate without good reason, PROSPERO administrators quickly raised the issue that multiple systematic reviews related to COVID-19, addressing the same research question, were being registered. Unnecessary duplication of research is wasteful and creates additional work for healthcare providers who need to make sense of evidence in a rapidly evolving field.[7–9] Duplication can also cause confusion, if findings between reviews are inconsistent.[2]

Using retrospective analyses, we aimed to explore the extent of duplication within PROSPERO registration records related to COVID-19, including whether the extent differed between rapid, full and living systematic reviews and/or particular areas of COVID-19-related research. We also aimed to explore the nature of duplication, including whether this was done unknowingly, covertly or for good reason by authors who were open and transparent about replication. Finally, we aimed to better understand the reasons for repetition from the authors' perspective and explore who funded duplicate reviews by conducting a survey.

## METHODS
### Searches
PROSPERO was searched for registrations relating to COVID-19 from March 2020 to January 2021. PROSPERO created a search strategy for users to identify COVID-19-related records, which was used in this research project (see online supplemental appendix A). When registering their review, authors chose their 'type and method of review' from a list. These tags then acted as filters on the PROSPERO website to identify the eligible studies. Due to resource constraints, not all research areas could be assessed. Four of the 17 COVID-19 research categories were selected for further study: epidemiology, transmission, treatment and rehabilitation. These research categories were chosen to capture a wide range of different topic areas addressing questions relating to COVID-19. For practical reasons, the records were downloaded for reviews of transmission, rehabilitation and epidemiology in November 2020, and treatment in January 2021.

## Identification of duplicate records

Within each topic area, records were each assigned a keyword based on the review title. Those with the same keyword were compared, first using the title of the registration record, and then for those with similar titles, using the Population, Intervention/Exposure, Comparator, Outcome, Study Design (PICOS) criteria recorded in the PROSPERO registration form. Those addressing the same research question, based on the PICOS criteria, were retained (see figure 2). To ensure that decisions were being made consistently, this was conducted in duplicate by two reviewers for a subset of records (the rehabilitation and transmission reviews, and a random sample of epidemiology registrations (100 from 445 records)). Discrepancies in decisions were discussed between reviewers to reach a consensus. The remaining records were screened singularly.

For reviews addressing the same research question, the date of first submission to PROSPERO and the date of publication were obtained. Registrations submitted after the publication of a record with a similar or identical PICOS were deemed 'duplicates' and retained. Reviews that have been submitted to PROSPERO but not yet registered/published on the public website (eg, if the record was returned to authors for additional information or clarification) are not visible on the PROSPERO COVID-19 search. Therefore, if a record was *submitted* before another record on the same topic and *published* after it, this was not counted as duplication because the authors would not have had knowledge of the other review at the point they submitted their registration form.

## Data extraction

For duplicate reviews, data on whether authors acknowledged similar existing registrations in PROSPERO when answering the screening questions prior to registration were recorded. From the public registration forms, data concerning the nature of duplication were extracted (see online supplemental appendix B for data extraction template). This included whether the authors provided reasons for conducting a new review and whether they had an external funding source. Furthermore, we extracted information about whether reviews were rapid, full or living systematic reviews. These topics were defined based on whether the PROSPERO record included the wording 'living' or 'rapid' in their record, if not they were deemed to be 'full' systematic reviews. This information was used to assess whether full systematic reviews were conducted after rapid reviews for particular research topics, and/or whether rapid or full systematic reviews were being registered after living systematic reviews.

## Surveys

To further understand the reasons why people duplicated existing reviews, authors of duplicate registrations were contacted using the email address of the named contact, provided in the PROSPERO form, and invited to complete a short questionnaire. Authors were asked whether they searched PROSPERO prior to registration, and whether they identified any similar reviews. Authors were also asked why they believed their review to be sufficiently different from an existing review, or why they felt repetition was needed if they had identified a similar review. The contacted authors had up to 6 weeks to respond to the survey, and reminders to complete the questionnaire were sent 1 week before it closed. A copy of the questionnaire is given in online supplemental appendix C.

## Patient and public involvement

Patients or the public were not involved in the design, conduct or reporting, or dissemination plans of our research. Our findings from this research were discussed with the COVID-19 Evidence Network to support Decision-making community.

## RESULTS
### Search results
From 1 March 2020 to 31 January 2021, PROSPERO registered 3013 protocols related to COVID-19. The monthly number of registrations published on PROSPERO peaked in April 2020 with 588 protocols (figure 1). Overall, 1054 registrations related to the four COVID-19 topic areas of interest (until January 2021).

### Descriptive characteristics
Five hundred and seventy-four records were found to have a similar review title and had their full PROSPERO record examined. One hundred and ninety-five of these records had a similar or identical PICO(S) (57 of which were the first instance of reviews addressing particular research questions). The remaining 138 duplicate records had been submitted when at least one protocol with similar or identical PICOS had already been published in PROSPERO. Figure 2 presents the number of records at each stage of the screening process. The data from the extracted duplicate studies are provided in online supplemental appendix D.

There were 33 (7.4% of those registered) duplicate reviews for epidemiological topics, 98 (20%) for treatments and 7 (7.5%) relating to the transmission of COVID-19. No duplicate reviews were identified in the rehabilitation topic area. Figure 3 shows the number of duplicate reviews registered by month between March 2020 and January 2021 in each research area.

The number of duplicate reviews on specific questions ranged from 1 to 14. Multiple duplicate registrations were most common in reviews evaluating COVID-19 treatments, including reviews evaluating the efficacy of drugs such as hydroxychloroquine (14 duplicate reviews), tocilizumab (7 duplicate reviews), remdesivir (6 duplicate reviews), the use of blood plasma (7 duplicate reviews) and prone position (6 duplicate reviews). The majority of epidemiological reviews only had a single duplicate review, except for those evaluating the impact of comorbidities (six duplicate reviews) and reviews of general

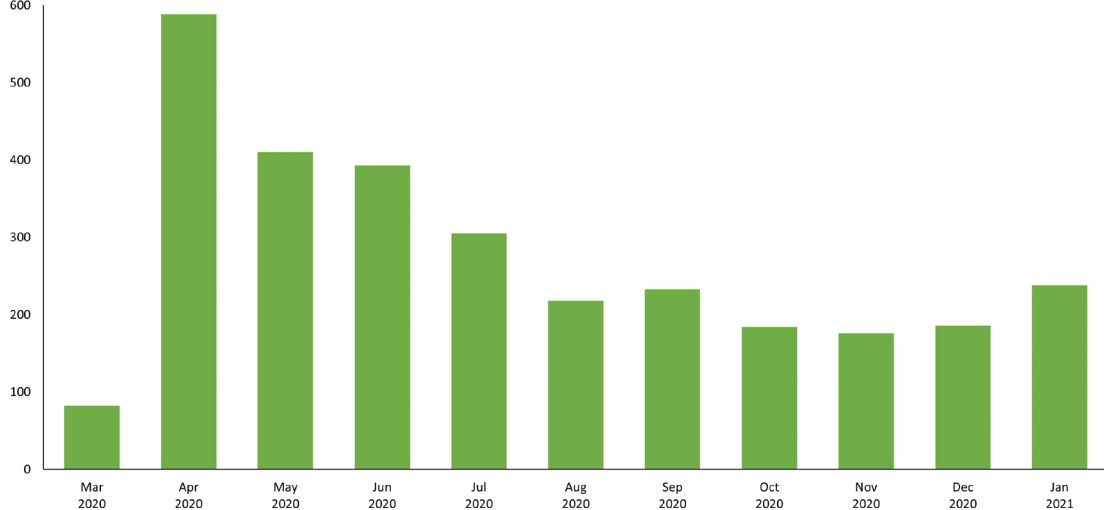

**Figure 1**   Number of COVID-19-related registrations between March 2020 and January 2021.

risk factors in patients with COVID-19 (eight duplicate reviews). The number of duplicates per research question is given in the online supplemental appendix E.

### Screening questions

When answering the PROSPERO screening questions prior to registering their protocol, authors of 85 of the 138 records considered as duplicate reviews did not acknowledge existing similar reviews in PROSPERO and selected 'Not Similar' on the screening question. Forty-eight indicated that their review was sufficiently different from those already registered and five authors indicated that repetition of existing reviews was needed.

### Review type

Table 1 shows the number of registrations published on PROSPERO by research area and review type. We identified 138 duplicate full systematic reviews, only two of which were replacing rapid reviews that addressed the same research question.

It was often difficult to differentiate between rapid and full systematic reviews as many of the registration forms were poorly completed. Some reviews also indicated they would complete within 4–6 weeks but did not identify as being a rapid review or detail abbreviated or rapid review methods, such as use of text mining during the screening process.

Twenty-three full systematic reviews and one living systematic review were found to be registered after living systematic reviews addressing the same research question.

### Reasons for duplication
#### Provided in registrations

For each duplicate record, we explored whether the authors had acknowledged previous registrations in their PROSPERO record, and if so, gave reasons for duplication. Thirty-three (24%) authors of duplicate records did so. Only four of these authors (12%) provided a reason for replication within their record despite being advised

during the registration process that the reasons for duplication should be made clear.

#### Provided in response to survey

We received 41 unique responses to the survey investigating reasons for duplication from 12 countries (figure 4). Eighteen responses (21% of those contacted) were from authors who, during PROSPERO screening, indicated that their review was 'not similar' to existing registered reviews, 21 responses (45% of those contacted) indicated their review was 'sufficiently different' and two responses (40% of those contacted) indicated 'repetition was needed'.

Of the respondents who provided their registration number in the survey, 68% had registered reviews relating to treatments for COVID-19. Overall, 90% of respondents had conducted a systematic review previously, and 92.5% intended to publish the results of their review. Most respondents had received no specific funding for their systematic review (87.5%).

Of the 18 respondents who indicated their review was 'not similar' to existing registered reviews, eight responded in the survey that they searched PROSPERO and did not identify any reviews on the same topic area. When asked to provide search terms used to do this, some were so broad that a large number of results would have been returned (such as single words like 'COVID-19'). Other respondents used multiple search terms; when we repeated these searches, all identified the duplicate records related to their review. Ten respondents said they searched PROSPERO and found some reviews on the same topics, but decided their review was different from the existing reviews found by their search.

All 21 respondents who indicated in the screening questions that their reviews were sufficiently different from other registered reviews responded that their searches identified some or all the reviews that we identified as being similar to their own. However, no respondent provided justification to why they believed their review

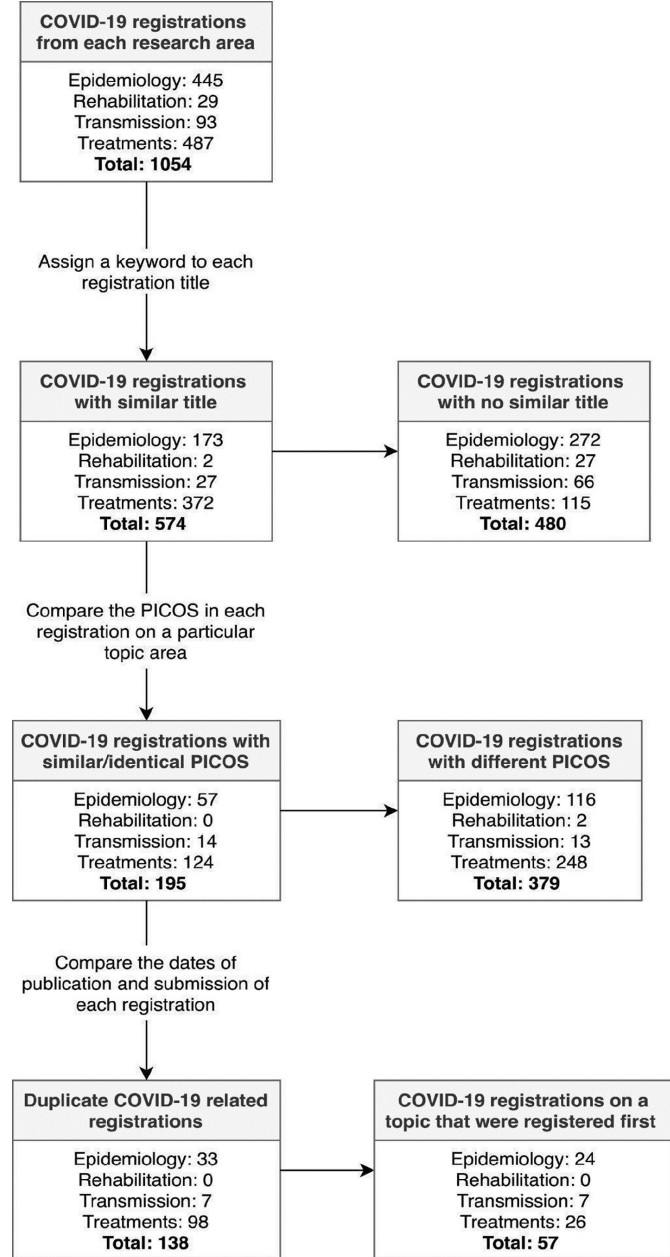

**Figure 2** Flow chart showing number of records screened and identified as 'duplicates' in the four COVID-19 research areas. PICOS, Population, Intervention/Exposure, Comparator, Outcome, Study Design.

was significantly different from previous registrations in their PROSPERO registration.

Of the two respondents who indicated that repetition was needed, neither provided their justification for repeating previous systematic reviews.

The reasons given for conducting similar reviews are summarised in table 2.

## DISCUSSION

To synthesise the considerable amount of primary research conducted during the COVID-19 pandemic,[7] a substantial number of systematic reviews have been

produced, many of which have been prospectively registered on PROSPERO. There is growing concern that these include duplicate systematic reviews that have replicated without good reason. Unintended or unnecessary duplication of research wastes research funding, time and effort and creates extra work for policy makers and healthcare providers who need to determine what novel information, if any, new reviews provide.[2 5 10]

We aimed to assess the extent and nature of duplication within PROSPERO registration records related to COVID-19, within four research areas: epidemiology, rehabilitation, transmission and treatment. Duplication was found to be extensive (accounting for 13% of records across these research areas). Researchers may have instead made a more valuable contribution to knowledge by considering alternative review topics (as advised on the PROSPERO home page).

The nature of this duplication was first examined by assessing whether duplication was a result of rapid reviews (a form of evidence synthesis in which some systematic review methods are condensed or omitted to produce results in a timely manner) being replaced by full systematic reviews.[11] As rapid reviews provide a useful overview of the available research but may stop short of providing a full or detailed synthesis, a later full systematic review may be justified. However, only two examples of this were identified in our sample. We also identified rapid and full systematic reviews registered after living systematic reviews on the same topic. For example, following the registration of a living systematic review and network meta-analysis on all interventions for COVID-19, an additional seven duplicate standard systematic reviews were registered. This is particularly wasteful as living reviews update regularly to capture and synthesise new research, preventing the need for many systematic reviews assessing the same research question at different points in time. Living reviews are particularly important in situations, such as the COVID-19 pandemic, where the research evolves rapidly and published standard systematic reviews may become out of date quickly.[12]

Responses to our survey provided more insight into the reasons for duplication. Notably, users indicated that they did not identify any similar records when they searched PROSPERO prior to registration, despite our own simple repetition of their searches identifying similar reviews. This may indicate poor use of search terms that are not able or likely to identify similar protocols, or lack of examination of the reviews returned by their search (the very broad terms that some authors reported using would have returned a large number of potentially similar reviews that could have made screening a daunting task). To tackle this, PROSPERO may need to develop better tools and guidance to support more effective searching and continue to promote research integrity, such as making explanation of planned replication a mandatory requirement.

While some valid reasons for duplication of systematic reviews were provided in the survey, for example,

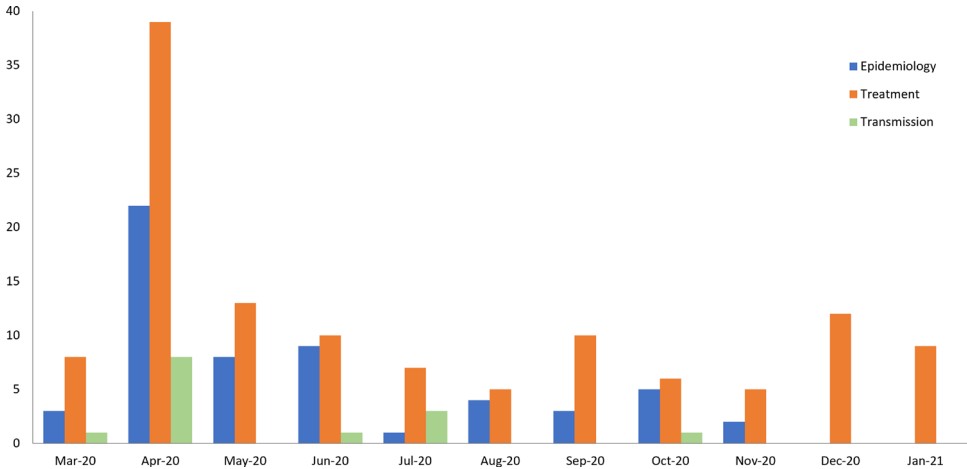

**Figure 3** Number of 'duplicate' reviews registered in the International Prospective Register of Systematic Reviews (PROSPERO) between March 2020 and January 2021 by research area. Registrations for transmission and epidemiology were obtained in November 2020, registrations on treatment were obtained in January 2021. Therefore, in December 2020 and January 2021, only records relating to treatment of COVID-19. No duplicate records relating to transmission were identified.

difference in the PICOS criteria and poor quality of previous reviews, these explanations were not included in the public PROSPERO registration forms, and therefore this detail can not inform decisions in future research, for example, umbrella reviews that synthesise outcome data from these systematic reviews.

Given the time and resources required to conduct systematic reviews, a surprising number of duplicate reviews lacked external funding. Unfunded reviews may have been initiated due to an altruistic desire to do something to help, and at least early in the pandemic, many funders were not set up to make rapid funding decisions. The number of reviews lacking external funding may, however, be concerning, at least in the case in high-income countries, as these reviewers may not have been subject to the same-level scrutiny or review as afforded by the funding application process (eg, demonstrating sufficient experience and a track record in conducting

systematic reviews). Research funders may also require due diligence in checking for existing reviews. Research that assesses whether funded proposals translate into better quality research, compared with self-funded proposals, could be useful.

Although not formally assessed here, a general observation from this research was the low quality of registration forms we reviewed. Poorly completed forms make it difficult, even for experienced systematic reviewers, to determine whether an existing review is similar or identical to their own and may therefore be contributing to unnecessary duplication of systematic reviews. Fewer high-quality systematic reviews are always more valuable to decision makers than many of poor-quality ones, which have the potential to damage the confidence of the research community in systematic reviews, which are viewed as the 'gold standard' of evidence-based research.[13 14]

**Table 1** Number of registered and duplicate reviews in four COVID-19 topic areas by type of systematic review

| Review type | Topic areas | | | | |
| --- | --- | --- | --- | --- | --- |
| | Epidemiology | Rehabilitation | Treatment | Transmission | Total |
| Registered reviews (n) | | | | | |
| Rapid | 16* | 2 | 21† | 7 | 46 |
| Living | 12 | 3 | 27 | 7 | 49 |
| Full | 417 | 24 | 439 | 79 | 520 |
| Total | 445 | 29 | 487 | 93 | 615 |
| Duplicate reviews (n) | | | | | |
| Rapid | 1 | 0 | 2 | 0 | 3 |
| Living | 1 | 0 | 6 | 1 | 8 |
| Full | 31 | 0 | 90 | 6 | 127 |
| Total | 33 | 0 | 98 | 7 | 138 |

*Two reviews were rapid, living systematic reviews and are counted in the 'living' row.
†One review was a rapid, living systematic review and is counted in the 'living' row.

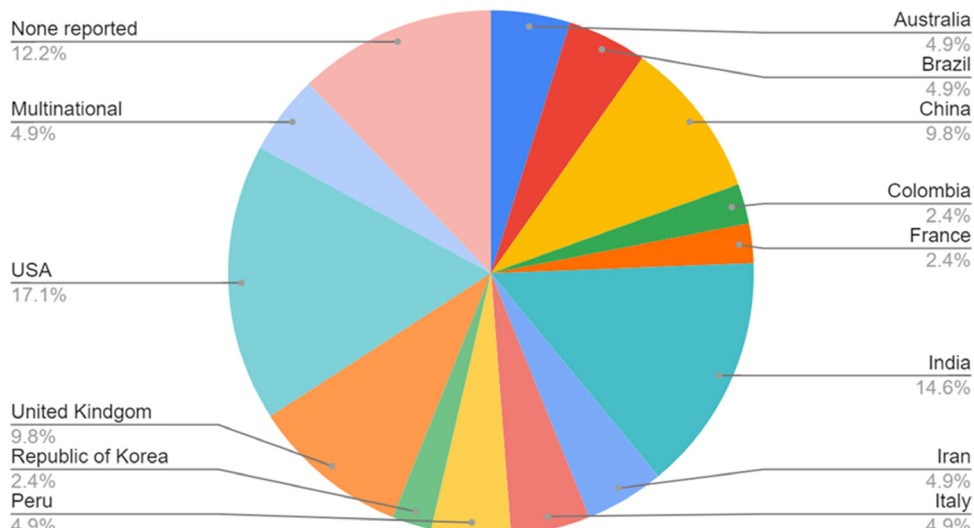

**Figure 4** The country in which the survey respondent's review is being carried out.

Journals may play a role in preventing systematic reviews that have been unnecessarily duplicated from being published. However, while some journals now require PROSPERO registration as a prerequisite for publication, peer reviewers or editors may not have the resource, nor deem it necessary to search PROSPERO for similar reviews to ascertain whether there is a need for a new systematic review. Issues around the low quality of registration forms, as previously discussed, may also impede the ability to do this and inform the editorial decisions. Further research may involve conducting a search for published reviews arising from the duplicate protocols identified in this research, to better understand how many make it to publication.

PROSPERO could help tackle the concerns raised in this research by engaging in outreach activity to improve the quality of registration records and by enhancing/promoting the guidance provided on the PROSPERO website. Further developments aimed at reducing the level of duplication could involve the provision of resources to help users decide whether duplication is

necessary or useful, such as the checklist created by Tugwell *et al.*[15] This could include information on the purpose of living systematic reviews and why duplication might be unnecessary after a research question has been addressed with this methodology. This advice would be especially helpful for researchers who lack experience in systematic reviews and may not understand the importance of avoiding duplication. An improved functionality around discovery of similar reviews could be introduced, including the development of automation that would highlight potentially similar reviews that exist in the database. If similar reviews are identified, additional information from the author could be requested, including the purpose and justification for duplication and the added value of their systematic review. More generally, PROSPERO's educational role could be developed in promoting best review methods and practice and in highlighting the importance of avoiding unintended duplication and research waste. With 175 140 signed up users (setting up a PROSPERO account is prerequisite to registering a review), we are well placed to reach a large audience and

| Table 2 | Summary of reasons for duplications from respondents of the questionnaire |
|---|---|
| **Response to screening question** | **Responses in survey** |
| Not similar | ▶ Different objectives or planned analyses.<br>▶ Poor quality of previous systematic reviews.<br>▶ Duplication was needed owing to the rapid rate that new studies on COVID-19 were being published. |
| Sufficiently different | ▶ Differences in PICOS, study design or planned analyses.<br>▶ Differences in databases searched.<br>▶ Poor quality of previous systematic reviews.<br>▶ Duplication was needed owing to the rapid rate that new studies on COVID-19 were being published. |
| Repetition needed | ▶ No systematic review was published or completed at the time of registration.<br>▶ Repetition was needed to confirm/refute previous systematic review conclusions.* |
| *Neither respondent provided reasons in their registration record.<br>PICOS, Population, Intervention/Exposure, Comparator, Outcome, Study Design. | |

help build expertise and infrastructure that would better serve decision-making in a future pandemic.

## Limitations

As not all authors prospectively register their systematic reviews, the real extent of duplication may differ from that in PROSPERO, particularly because those who knowingly duplicate without good reason may be less likely to register their review.

Owing to time and resource constraints our research only examined four of the 17 research areas related to COVID-19. Although it is not our intention to repeat this research for the remaining research categories, this could be an interesting area for future research, particularly to see whether duplication remains consistently apparent in more recent areas of research related to COVID-19, for example, Long COVID.

As records can be created in the PROSPERO system prior to submission, it is possible that similar systematic reviews could be registered during this period. If authors did not re-search PROSPERO prior to submission, unintended duplication could occur. However, these data are not available in the PROSPERO system so were not captured in this study.

We observed low-quality registration forms and a substantial number of systematic reviews that lacked external funding; however, a comparison was not made to non-duplicate reviews. Further work addressing this could be of interest.

Finally, while the surveys provide a useful insight into the reasons for replication of previous systematic reviews, the number of responses received was small (41 responses/138 contacted). This may not be representative of all PROSPERO users, as those who responded could feel that they had a legitimate reason to replicate, while those who did not may not have responded to the survey when asked.

## CONCLUSIONS

This research has highlighted that registration of duplicate systematic reviews related to COVID-19 has occurred frequently in PROSPERO, with 138 review teams addressing research questions of existing reviews in the first year of the pandemic. It adds to the evidence highlighting the problem of duplication and research waste in reviews and evidence syntheses and has identified a need to increase awareness of the need to avoid unintended duplication among those conducting reviews. PROSPERO, which has provided registration free of charge and served the systematic review community for 10 years, responded to the COVID-19 pandemic by ensuring that reviews relating to COVID-19 were prioritised and registered/published rapidly. Changes made to accommodate this, along with associated reflection on the role of registration, have informed thinking about future developments and consideration of how PROSPERO can continue to support the global systematic review community over the next 10 years and be part of the research infrastructure that stands ready to respond effectively to future global health challenges.

**Acknowledgements** We would like to acknowledge Gordon Dooley who extracted the data from PROSPERO for analysis.

**Contributors** LB and RW equally contributed to the data curation, formal analysis, investigation, visualisation and writing of the original draft. RW is responsible for the overall content as the guarantor. LS provided support for these tasks. LS led the conceptualisation, project administration, validation, and reviewing and editing of the drafts. LB and RW provided support for these tasks. All authors contributed equally to the methodology used in the research. LS provided overall project supervision..

**Funding** This study/project is funded by the NIHR Technology Assessment Reviews Programme (14/25/13).

**Disclaimer** This report presents independent research commissioned by the NIHR. The views and opinions expressed in this publication are those of the authors and do not necessarily reflect those of the NHS, the NIHR, MRC, CCF, NETSCC, Health Technology Assessment Programme or the Department of Health and Social Care.

**Competing interests** Production of PROSPERO is funded by the NIHR through an overarching research contract 14/25/13. Supplementary funding was made through this award to support PROSPERO's response to COVID-19 which included this project. This research funding is made to the University of York. LS is the principal investigator and award holder. RW is partially funded on this grant and is part of the PROSPERO team. LB was funded by the supplementary COVID funding for a period of 6 months to provide additional support to PROSPERO and contribute to this project.

**Patient and public involvement** Patients and/or the public were not involved in the design, or conduct, or reporting, or dissemination plans of this research.

**Patient consent for publication** Not applicable.

**Ethics approval** The study was granted ethical approval from the University of York, Health Sciences Research Governance Committee.

**Provenance and peer review** Not commissioned; externally peer reviewed.

**Data availability statement** Data are available upon reasonable request. The data that support the findings of this study are available from the corresponding author upon reasonable request.

**ORCID iD**
Lucy Beresford http://orcid.org/0000-0001-6803-5566

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
