## [Reviewer comments · BMJ Open]

ARTICLE DETAILS

TITLE (PROVISIONAL)	The extent and nature of duplication in PROSPERO using COVID-19 related registrations: a retrospective investigation and survey
AUTHORS	Beresford, Lucy; Walker, Ruth; Stewart, Lesley

VERSION 1 – REVIEW

REVIEWER	Robert F Terry WHO, TDR
REVIEW RETURNED	01-Apr-2022

GENERAL COMMENTS	In the Methods section it would be useful to reference Figure 2 and include the actual search terms used to search PROSPERO. If this is the purpose of the table in the Appendix this just needs further clarification in the Methods section. The authors state data sharing is not applicable. However, it would be useful to deposit the dataset as a supplementary file here or in a suitable repository as a list of the full article citation details that were analysed so that a follow up study looking at how many of these protocols were completed and published could be easily undertaken. Clearly duplication of protocols is a waste of valuable research time but the majority of these protocols were not specifically funded to undertake these reviews indicating the researchers were undertaking these studies within their existing research agenda/funds. Therefore, what is the real lost opportunity costs here? The key issue for users, such as policy makers, would be in the published results. Therefore, I think the major issue here is how many of these protocols do go on to be published and whether 'duplication' is weeded out of journals? The discussion might want to elaborate on the role of journals and if they use the registration of protocols to make editorial decisions on publication. This suggestion to analyse publication rates could be added to the discussion as further research or, if the authors have the capacity, a search for published reviews arising from these protocols could be undertaken as additional research within this study. Again, if capacity exists, I would be interested to know the response of the PROSPERO administrators to these findings either directly or through a peer review of this paper. This is a good paper that highlights the value of monitoring research but also exposes its limitations when data entry is poor and essentially we have self-regulation to reduce duplicates without any meaningful consequences when duplicate research is shown.
---

REVIEWER	Amanda Blatch-Jones National Institute for Health Research Evaluation, Trials and Studies Coordinating Centre (NETSCC), University of Southampton
REVIEW RETURNED	25-Apr-2022

GENERAL COMMENTS	Overall, this is a well written manuscript and the authors have clearly provided sufficient background to why this study was needed and conducted. However, this isn't reflected in the abstract, so I would recommend that an introduction/background section is included in the abstract (as noted below). I've provided some minor comments under each of the following headings: Abstract: There is no introduction section in the abstract to allow the reader to determine why and for what reason this study is needed/what the current landscape is/gaps in the evidence. Under the results how many people were contacted (was the response rate out of 41? For example) and was it the corresponding author who was contacted? Is it worth noting the design of the study in the abstract to help the reader understand, at a glance, how the study was conducted? What were the number of responses against the reasons for duplication? I feel the abstract needs more detail and provide the same quality as the manuscript (which is well written and thorough). Introduction: The first use of PROSPERO needs to be fully abbreviated (line 14). The first use of NIHR also is required after it has been fully abbreviated (line 16). Also note that the NIHR has recently changed its abbreviation to the National Institute for Health and Care Research (NIHR) so this will need to be reflected in the manuscript. The authors have provided sufficient references and justification for why this study was needed and puts the study into perspective, around the waste in research and avoidable waste agenda (needs to be reflected in the abstract). However, more up to date references could be included to strengthen the case about issues around duplication and the need for transparent reporting when registering a review/study etc. It might be worth expanding the paragraph about PROSPERO, for readers who are not familiar with it (in terms of records, value, scope and its success over the last 10 years). The last paragraph in the introduction requires some evidence to back up the authors claims around unnecessary duplication and its potential impact on research and the wider society. I would suggest that the last sentences around the aim of the paper are a separate paragraph to make it easier for the reader to see what the aim, purpose and objective is and why this paper is needed. Methods: What is the design of the study? How was PROSPERO searched for the relevant registrations (what key terms were used to search the database), who searched and were the records verified by someone else? More details around the four COVID-19 research categories is required. Are these categories researchers assign during the registration, or were these categories assigned for the purpose of this study? How were these categories chosen (are they from a prespecified classification, or not?) Were there other COVID-19 reviews that were excluded? And if so what were they, in terms of research category? Under data extraction, how was and what was the definition used to categorise rapid, full or living systematic review?
--

	For the survey, who what and when was the survey sent out? How long did they have to complete the survey before a reminder was sent? Were the questions of the survey piloted before sending out? Results: The methods state to January 2021, but the first sentence in the results sections says January 2022. What is the correct year of extraction of records? The figure defines the period March 2022 to July 2021. Why to July, when the period under review was March 2020 to January 2021. The details and the figure need to match, as it currently stands it does not match, and there is no reason/justification why the figure goes to July 2021. I would suggest only reporting data for the period the study was investigating - March 2020 to January 2021. The paragraph on rows 45-55, I'm assuming the numbers in brackets is referring to the number of reviews, I would make this clearer or change to percentages so that it is consistent with the previous paragraph (if less than 10 the units need to be written out in full, as per format throughout the manuscript). 47 needs to be written out. Table 1 - it would be helpful to also include the Totals for rows and columns. In the survey results - how many invitations were sent out by email? When was the invitation sent out? A response rate could then be provided for the reader. I would suggest considering the 11/19 and 8/19 sentences and possibly revise to be "Of the 11 responses who indicated...." "Eight respondents said they....." (possibly repeat for the other sentences starting in the same way). Discussion: Possible reference error on line 28. Although the study found duplication, as stated in the opening paragraph of the discussion, the survey responses seem to suggest that the reasons for another review was due to poor quality and differences in the PICOS. Were these reasons and those found in the PROSPERO database matched up, to justify the duplication? So although duplication exists the reasoning for the slight variation does matter, as it may not be used in future systematic reviews, meta analyses and/or Cochrane reviews (due to criteria). The authors pick up on a valid point, around living systematic reviews and there intended purpose, whereby the duplication is particularly wasteful given the scope and purpose of living systematic reviews. I feel this could be brought out more in the conclusion, for future consideration and recommendation for PROSPERO, funders and researchers. The discussion is clearly written and covers many outcomes arising from the study. These are clearly evidenced as recommendations for which researchers and PROSPERO need to consider in the future. One of the limitations mentioned is the 4 out of 17 research categories (a point raised under methods). It would be good to include this in the methods section to ensure transparency for the reader and to put the results in some context - only a quarter of the research categories were assessed due to resource and time constraints. Given the findings of this study is there any intention to repeat this study for the remaining categories? For the survey section under limitations, including the response rate (as mentioned above) would be useful for the reader (again thinking about transparency, but also putting the study into context and demonstrating the challenge of surveys during a pandemic etc). Under the conclusions, would it also be viable that more details from authors when duplication is reported, is requested? The purpose for the duplication, justification for the duplication, and the value adding aspect of the duplication (in some cases duplication may be required due to different health care systems, quality of previous reviews, risk of bias assessment requirements, criteria assessment, populations
--	--

	etc). Funding statement: I would recommend taking our CCF and NETSCC as these are local abbreviations for two coordinating centres (readers may not know the difference, and in most instances do not depict between centres, they only see NIHR). As the authors have not stated what these two abbreviations are, I would recommend only using NIHR (note above comment, that it is now the National Institute for Health and Care Research). Data Sharing statement: I'm not clear why this is not applicable. Data from the extraction of PROSPERO and survey data from the respondents have arisen as part of the study. How, on request, can data be shared with others? Do you have a data sharing policy? PROSPERO data would be in the public domain, so how would data from the survey protect individuals' responses (the survey states that responses are completely anonymised)? References: Although the authors have provided sufficient references, there is a suite of papers around waste in research, which may be appropriate to include. Also, more current papers around the value of registries and the need to register studies (and reviews) have been / are a topic of debate. It might be worthwhile including some of these too? Finally, I would suggest that the appendices are labelled either numerically or alphabetically so that the reader can find the relevant appendix easily. In summary, this is a well written paper covering an important area for future consideration, not only for registries, but also for funders and researchers conducting reviews. The authors raise some important points which could vastly reduce duplication of effort and unnecessary waste of research time and funding from research organisations.
--	---

VERSION 1 – AUTHOR RESPONSE

Reviewer's Feedback		Author's Comments
Dr Robert F Terry, WHO		
Methods	In the Methods section it would be useful to reference Figure 2 and include the actual search terms used to search PROSPERO. If this is the purpose of the table in the Appendix this just needs further clarification in the Methods section.	The searches paragraph has been expanded to provide details of the search terms used to identify COVID-19 related records, and details of how we obtained the records for each topic area. “PROSPERO was searched for registrations relating to COVID-19 from March 2020 to January 2021. PROSPERO created a search strategy for users to identify COVID-19 related records, which was used in this research project (see Appendix A). Four of the seventeen COVID-19 research categories were selected for further study: epidemiology, transmission, treatment, and rehabilitation. When registering their review, authors chose their ‘type and method of review’ from a list. These tags then acted as filters on the PROSPERO website to identify the eligible

		studies” Figure 2 has been referenced in the Methods section (under Identification of Duplicate Records).
Results	The authors state data sharing is not applicable. However, it would be useful to deposit the dataset as a supplementary file here or in a suitable repository as a list of the full article citation details that were analysed so that a follow up study looking at how many of these protocols were completed and published could be easily undertaken.	The data from the extracted duplicate studies have been provided in Appendix D.
Discussion	The discussion might want to elaborate on the role of journals and if they use the registration of protocols to make editorial decisions on publication. This suggestion to analyse publication rates could be added to the discussion as further research or, if the authors have the capacity, a search for published reviews arising from these protocols could be undertaken as additional research within this study.	Thank you for this comment. A paragraph has been added the discussion of this paper to elaborate on the role of journals in preventing duplication of systematic reviews: “Journals may play a role in preventing systematic reviews that have been unnecessarily duplicated from being published. However, whilst some journals now require PROSPERO registration as a prerequisite for publication, peer reviewers or editors may not have the resource, nor deem it necessary to search PROSPERO for similar reviews to ascertain whether there is a need for a new systematic review. Issues around the low quality of registration forms, as previously discussed, may also impede the ability to do this and inform the editorial decisions. Further research may involve conducting a search for published reviews arising from the duplicated protocols identified in this research, to better understand how many make it to publication.”
	I would be interested to know the response of the PROSPERO administrators to these findings either directly or through a peer review of this paper.	Ruth Walker acts as scientific lead on PROSPERO and is involved in the day to day running of the database, as well as processing submissions received to the database. Lucy Bresford has previously worked processing submissions received to the database. It was the PROSPERO administrators who firstly noted the large number of systematic reviews addressing similar research questions. This has been made more explicit in the methods section

		of the paper: “Despite encouragement to check PROSPERO for similar systematic reviews and to not duplicate without good reason, PROSPERO administrators quickly raised the issue that multiple systematic reviews related to COVID-19, addressing the same research question were being registered.”
Ms. Amanda Blatch-Jones, National Institute for Health Research Evaluation, Trials and Studies Coordinating Centre (NETSCC)		
Abstract	There is no introduction section in the abstract to allow the reader to determine why and for what reason this study is needed/what the current landscape is/gaps in the evidence.	An introductory paragraph has been added. “During COVID-19, PROSPERO (an international prospective register of systematic reviews) experienced a surge in registrations for COVID-19 related systematic reviews, and duplication of research questions became apparent. Duplication can waste funding, time and research effort and can make policy-making more difficult.”
	Under the results how many people were contacted (was the response rate out of 41? For example) and was it the corresponding author who was contacted?	We have clarified in the abstract: “The named contact authors of ‘duplicate’ registrations were invited to complete a short survey, asking whether they searched PROSPERO prior to registration, identified the similar reviews, and if so, why they continued with their review.” “From the 138 authors invited to take part in the short survey, we received 41 responses..” In the methods section we have clarified: “..authors of duplicate registrations were contacted using the email address of the named contact, provided in the PROSPERO form, and invited to complete a short questionnaire.”
	Is it worth noting the design of the study in the abstract to help the reader understand, at a glance, how the study was conducted?	We have updated the text in the abstract to expand on the methods used: “We searched PROSPERO for registrations

		related to four COVID-19 research areas: epidemiology, rehabilitation, transmission and treatments. Records identified were compared using the PICOS (Population, Intervention, Comparator, Study Design) elements of the PROSPERO registration form. Registrations that had similar or identical PICOS were retained for further evaluation as ‘duplicates’.
	What were the number of responses against the reasons for duplication?	The number of responses against the reasons for duplication have been added in.
Introduction	The first use of PROSPERO needs to be fully abbreviated (line 14).	PROSPERO is not abbreviated and doesn't stand for anything, so no changes to this were made.
	The first use of NIHR also is required after it has been fully abbreviated (line 16). Also note that the NIHR has recently changed its abbreviation to the National Institute for Health and Care Research (NIHR) so this will need to be reflected in the manuscript.	The full term for the NIHR (to include 'care') has been added, and an abbreviation has been included.
	It might be worth expanding the paragraph about PROSPERO, for readers who are not familiar with it (in terms of records, value, scope and its success over the last 10 years).	We have added additional text highlighting the value and success of PROSPERO since its launch: “In the ten years since its launch in 2011, prospective registration of systematic reviews has become routine and the importance of this, acknowledged widely. At launch PROSPERO anticipated registration of around 2000 systematic reviews per year. In 2020 40,639 reviews were registered, and PROSPERO now includes over 125,000 registration records. PROSPERO has global reach, for example, in 2020 registrations originated from 194 countries and territories.”
	The last paragraph in the introduction requires some evidence to back up the author's claims around unnecessary duplication and its potential impact on research and the wider society.	Additional references have been added to this section.

	I would suggest that the last sentences around the aim of the paper are a separate paragraph to make it easier for the reader to see what the aim, purpose and objective is and why this paper is needed.	Thank you, this has been edited to have a separate paragraph.
Methods	What is the design of the study?	The study designs used have been added to the final paragraph of the introduction. “Using retrospective analyses, We aimed to explore the extent of duplication within PROSPERO registration records related to COVID-19” “Finally, we aimed to better understand the reasons for repetition from the authors’ perspective and explore who funded duplicate reviews by conducting a survey.”
	How was PROSPERO searched for the relevant registrations (what key terms were used to search the database), who searched and were the records verified by someone else?	The searches paragraph has been expanded to provide details of the search terms used to identify COVID-19 related records, and details of how we obtained the records for each topic area. “PROSPERO was searched for registrations relating to COVID-19 from March 2020 to January 2021. PROSPERO created a search strategy for users to identify COVID-19 related records, which was used in this research project (see Appendix A). Four of the seventeen COVID-19 research categories were selected for further study: epidemiology, transmission, treatment, and rehabilitation. When registering their review, authors chose their ‘type and method of review’ from a list. These tags then acted as filters on the PROSPERO website to identify the eligible studies.”
	More details around the four COVID-19 research categories is required. Are these categories researchers assigned during the registration, or were these categories assigned for the purpose of this study? How were these categories chosen (are they from a prespecified classification, or not?) Were there other COVID-19 reviews that were excluded?	This sentence has been amended to provide justification for only choosing four topic areas. “Four of the seventeen COVID-19 research categories were selected for further study: epidemiology, transmission, treatment, and rehabilitation; due to resource constraints, not all research areas could be assessed. These research categories were chosen to capture a wide range of different topic areas addressing

	And if so what were they, in terms of research category?	questions relating to COVID-19.”
	Under data extraction, how was and what was the definition used to categorise rapid, full or living systematic review?	Further information regarding the categorisation of how living, full and rapid reviews were defined has been added in: ‘Furthermore, we extracted information about whether reviews were rapid, full, or living systematic reviews. These topics were defined based on whether the PROSPERO record included the wording ‘living’ or ‘rapid’ in their record, if not they were deemed to be ‘full’ systematic reviews.’
	For the survey, who, what and when was the survey sent out? How long did they have to complete the survey before a reminder was sent? Were the questions of the survey piloted before sending out?	Additional information has been added to the methods section to provide more information about who the survey was sent to, and how long it remained open for. “To further understand the reasons why people duplicated existing reviews, authors of duplicate registrations were contacted using the email address of the named contact, provided in the PROSPERO form” “The contacted authors had up to six weeks to respond to the survey, and reminders to complete the questionnaire were sent one week before it closed”
Results	The methods state to January 2021, but the first sentence in the results sections says January 2022. What is the correct year of extraction of records?	The first line of the sentence was providing an overall number of records that were registered on PROSPERO from the start of the pandemic up until the point where we submitted our research to BMJ Open. However, we understand that this could cause confusion, and so this has been amended to provide the total number of PROSPERO records registered from March 2020 to January 2021, when we finished searches in PROSPERO. The first sentence in the results section has been amended to ‘2021’.
	The figure defines the period March 2022 to July 2021. Why to July, when the period under review was March 2020 to January 2021. The details and the figure need to match, as it currently stands it does not match, and there is no reason/justification	Thank you for noticing this, we understand that there could be confusion in the discrepancy in the information provided. The figure has been amended to only included data from March 2020 to January 2021.

	why the figure goes to July 2021. I would suggest only reporting data for the period the study was investigating - March 2020 to January 2021.	
	The paragraph on rows 45-55, I'm assuming the numbers in brackets is referring to the number of reviews, I would make this clearer or change to percentages so that it is consistent with the previous paragraph (if less than 10 the units need to be written out in full, as per format throughout the manuscript).	This has been altered. I have altered the numbers to text (if they're under 10) and added 'duplicate reviews' in the brackets.
	47 needs to be written out.	This has been changed to text.
	Table 1 - it would be helpful to also include the Totals for rows and columns.	The totals have been added to Table 1
	I would suggest considering the 11/19 and 8/19 sentences and possibly revise to be "Of the 11 responses who indicated...." "Eight respondents said they....." (possibly repeat for the other sentences starting in the same way).	This has been amended.
Discussion	Possible reference error on line 28.	Thank you, this has been removed.
	Although the study found duplication, as stated in the opening paragraph of the discussion, the survey responses seem to suggest that the reasons for another review was due to poor quality and differences in the PICOS. Were these reasons and those found in the PROSPERO database matched up, to justify the duplication? So although duplication exists the reasoning for the slight variation does matter, as it may not be used in future systematic reviews, meta analyses and/or Cochrane reviews (due to criteria)	Thank you for this comment we have included a sentence in the discussion to address this: "Whilst some valid reasons for duplication of systematic reviews were provided in the survey, for example, difference in the PICOS criteria and poor quality of previous reviews, these explanations were not included in the public PROSPERO registration forms, and therefore this detail can not inform decisions in future research for example, umbrella reviews that that to synthesise outcome data from these systematic reviews."

	The authors pick up on a valid point, around living systematic reviews and there intended purpose, whereby the duplication is particularly wasteful given the scope and purpose of living systematic reviews. I feel this could be brought out more in the conclusion, for future consideration and recommendation for PROSPERO, funders and researchers	Thank you, this has now been addressed specifically in the concluding paragraph: “Further developments aimed at reducing the level of duplication could involve the provision of resources to help users decide whether duplication is necessary or useful, such as the checklist created by Tugwell et al (2021).¹¹ This could include information on the purpose of living systematic reviews and why duplication might be necessary after a research question has been addressed with this methodology.”
	One of the limitations mentioned is the 4 out of 17 research categories (a point raised under methods). It would be good to include this in the methods section to ensure transparency for the reader and to put the results in some context - only a quarter of the research categories were assessed due to resource and time constraints. Given the findings of this study is there any intention to repeat this study for the remaining categories?	Thank you for this. We had updated the methods section to say: “Four/seventeen COVID-19 research categories were selected for further study: epidemiology, transmission, treatment, and rehabilitation; due to resource constraints, not all research areas could be assessed.” We do not intend to repeat this research for the remaining categories but have suggested this as an area for future research in the limitations section: “Owing to time and resource constraints our research only examined four of the seventeen research areas related to COVID-19. Although it is not our intention to repeat this research for the remaining research categories, this could be an interesting area for future research, particularly to see whether duplication remains consistently apparent in more recent areas of research related to COVID-19 for example, long-covid. “
	For the survey section under limitations, including the response rate (as mentioned above) would be useful for the reader (again thinking about transparency, but also putting the study into context and demonstrating the challenge of surveys during a pandemic etc).	Thank you. The response rate has been added to the limitations section.
	Under the conclusions, would it also be viable that more details	Thank you. This has been added to the final

	from authors when duplication is reported, is requested? The purpose for the duplication, justification for the duplication, and the value adding aspect of the duplication (in some cases duplication may be required due to different health care systems, quality of previous reviews, risk of bias assessment requirements, criteria assessment, populations etc).	paragraph of the discussion section: “An improved functionality around discovery of similar reviews could be introduced, including the development of automation that would highlight potentially similar reviews that exist in the database. If similar reviews are identified, additional information from the author could be requested, including the purpose and justification for duplication and the added value of their systematic review.”
Funding statement	I would recommend taking our CCF and NETSCC as these are local abbreviations for two coordinating centres (readers may not know the difference, and in most instances do not depict between centres, they only see NIHR). As the authors have not stated what these two abbreviations are, I would recommend only using NIHR (note above comment, that it is now the National Institute for Health and Care Research).	We have updated the funding statement to reflect guidance provided by the NIHR.
Data Sharing statement	I'm not clear why this is not applicable. Data from the extraction of PROSPERO and survey data from the respondents have arisen as part of the study. How, on request, can data be shared with others? Do you have a data sharing policy? PROSPERO data would be in the public domain, so how would data from the survey protect individuals' responses (the survey states that responses are completely anonymised)?	We have addressed this to say the following: “Data from the extracted duplicated studies have been provided in Appendix D. Anonymised data from the survey can be obtained upon request.”
References	Although the authors have provided sufficient references, there is a suite of papers around waste in research, which may be appropriate to include. Also, more current papers around the value of registries and the need to register studies (and reviews) have been / are a topic of debate. It might be worthwhile including some of these too?	A number of papers have been added to the reference list to further support statements made in the discussion and introduction. This includes  • McDonald, S., et al. (2022). "Most published systematic reviews of remdesivir for COVID-19 were redundant and lacked currency." Journal of clinical epidemiology 146: 22-31. • Pieper, D. and T. Rombey (2022). "Where to prospectively register a

		systematic review." Systematic Reviews 11(1): 8.  • Siontis, K. C. and J. P. A. Ioannidis (2018). "Replication, Duplication, and Waste in a Quarter Million Systematic Reviews and Meta-Analyses." Circ Cardiovasc Qual Outcomes 11(12): e005212. • Helliwell, J. A., et al. (2022). "Duplication and nonregistration of COVID-19 systematic reviews: Bibliometric review." Health Science Reports 5(3): e541.
Appendix	Finally, I would suggest that the appendices are labelled either numerically or alphabetically so that the reader can find the relevant appendix easily.	This has been amended, so each section is separated and labelled alphabetically

VERSION 2 – REVIEW

REVIEWER	Robert F Terry WHO, TDR
REVIEW RETURNED	27-Jul-2022
GENERAL COMMENTS	Really interesting paper and a great research question. It will be interesting to see what action could be taken as a response to reduce this 'research waste'.